# Alternating Optimized Stochastic Vector Quantization in Neural Compression

## Abstract

In neural compression, vector quantization (VQ) is usually replaced by a differentiable approximation during training for gradient backpropagation. However, prior approximation methods face two main issues: 1) the train-test mismatch between differentiable approximation and actual quantization, and 2) the suboptimal encoder gradients for rate-distortion (RD) optimization. In this paper, we first provide new finds about how approximation methods influence the RD optimization in neural compression, and then propose a new solution based on these finds. Specifically, if a neural compressor is regarded as a source-space VQ, we find that the encoder implicitly determines the quantization boundaries, and the decoder determines the quantization centers. Suboptimal approximation methods lead to suboptimal gradients for RD optimization of quantization boundaries and centers. Therefore, to address the first issue, we propose an encode-decoder alternating optimization strategy. The encoder is optimized with differentiable approximation, and the decoder is optimized with actual quantization to avoid the train-test mismatch of quantization centers. To address the second issue, we propose a sphere-noise based stochastic approximation method. During encoder optimization, VQ is replaced with a uniform sphere noise centered at the input vector. When the input vector is located at the quantization boundary, the encoder gradient is closer to the difference in RD loss between adjacent quantization centers, facilitating better encoder optimization. We name the combination of optimization strategy and approximation method as Alternating Optimized Stochastic Vector Quantization. Experimental results on various vector sources and natural images demonstrate the effectiveness of our method.

## 1 Introduction

Quantization is a classical lossy compression technique. In theory, vector quantization (VQ) Gersho & Gray (1992) can achieve optimal rate-distortion (RD) performance in source coding. However, the exponentially increasing complexity of VQ and its non-differentiable nature limit its practical use in neural compression Ballé et al. (2017); Ballé et al. (2020); Lu et al. (2019); Li et al. (2021), particularly for high-dimensional data. The complexity issue can be addressed by simplifying VQ to scalar quantization Ballé et al. (2020), multistage VQ Feng et al. (2023); Zhu et al. (2022) or lattice VQ Zhang & Wu (2023). In this paper, we focus on tackling the non-differentiability issue of VQ for end-to-end RD optimization.

In neural compression, quantization is performed in the latent space of an autoencoder. Since quantization is non-differentiable, optimizing the learnable encoder transform presents a significant challenge. A typical solution is to introduce a differentiable approximation of quantization during training, such as additive uniform noise Ballé et al. (2017) and straight-through estimator (STE) Bengio et al. (2013). However, prior works mainly focus on a special case of VQ, *i.e.*, uniform scalar quantization. For general vector quantization, the optimization problem remains unresolved and primarily involves two issues. The first issue is train-test mismatch. Differentiable approximations Agustsson et al. (2017a); Zhu et al. (2022) often differ from actual quantization, resulting in mismatch in the decoder's reconstruction between training and testing. The second issue is the suboptimality of encoder gradients. Although previous approximation methods are differentiable, the gradients backpropagated to the encoder remain suboptimal under the RD criterion.

In this paper, we aim to design an VQ approximation method for end-to-end RD optimization in neural compression. Since the gradients for the encoder and decoder vary depending on the approximation method, the first step is to understand how they influence RD performance. By interpreting a neural compressor as a source-space vector quantizer, we show that the encoder function implicitly determines the quantization boundaries, and the decoder function determines the quantization centers. Suboptimal boundaries and centers directly lead to suboptimal RD performance in lossy compression. Thus, the encoder gradient at the boundaries and the decoder's gradient at the centers are key factors influencing compression performance. In theory, entropy-constrained vector quantization (ECVQ) Chou et al. (1989) has the optimal quantization boundaries and centers.

To address the train-test mismatch issue, we propose an encode-decoder alternating optimization strategy. When optimizing the quantization centers, the encoder is fixed, and the decoder and codebook are optimized using actual quantization. When optimizing the quantization boundaries, the decoder and codebook are fixed, and the encoder is optimized using the approximation method. These two steps alternate during training, ensuring consistent decoder reconstruction while allowing gradients to be backpropagated to the encoder.

To address the issue of suboptimal encoder gradients, we first provide gradient analysis and argue that prior approximation methods are suboptimal for RD performance due to two reasons: 1) discontinuous encoder gradients result in non-smooth quantization boundaries, and 2) the encoder gradients at boundaries should align with the RD loss differences when quantizing to nearby centers. In theoretically optimal ECVQ, the RD loss for an input vector at the quantization boundary is equal when quantized to the two neighboring centers. Therefore, if the encoder gradient at the boundary closely approximates the loss difference between neighboring centers, it will help the encoder learn better quantization boundaries. Based on this analysis, we propose a sphere-noise based stochastic approximation method. This quantization approximation follows a uniform sphere distribution centered at the input vector, with the radius of the hypersphere equal to the distance between the input vector and the nearest quantization center. We further demonstrate that the encoder gradient is equivalent to the integral of the RD function over the surface of the high-dimensional sphere. When the input vector lies at the quantization boundary, the gradient is closer to the difference in RD loss between adjacent quantization centers, leading to more effective encoder optimization.

By combining the proposed alternating optimization strategy and shere-noise based stochas- tic approximation, we propose a new method named Alternating Optimized Stochastic Vector Quantization for end-to-end RD optimization. We provide comprehensive experiments and analysis on various vector sources. Experimental results on neural image compression further demonstrate the effectiveness of the proposed method.

## 2 RELATED WORK

Most existing works in neural compression follows the structure of nonlinear transform coding Ballé et al. (2020), with a pair of learnable transform, an entropy model and a vector quantizer in latent space. As shown in Figure 1, the encoder transform $g_a$ maps the input vector $\boldsymbol{x}$ into latent vector $\boldsymbol{y}$, which is then quantized by a quantizer $\hat{\boldsymbol{y}} = Q_y(\boldsymbol{y}) = Q_y^d(Q_y^e(y))$. $Q_y^e$ is the quantization encoder that maps $\boldsymbol{y}$ to discrete index $\boldsymbol{i}$, and the quantization decoder $Q_y^d$ maps $\boldsymbol{i}$ to quantized vector $\hat{\boldsymbol{y}}$. The entropy model $p_{\boldsymbol{i}}$ is used to model the distribution of index $\boldsymbol{i}$ for entropy coding. The optimization target is to minimize the RD loss $L = R + \lambda D$, where $R = \mathbb{E}_{\boldsymbol{x}}[-\log p_{\boldsymbol{i}}(\boldsymbol{i})] = \mathbb{E}_{\boldsymbol{x}}[-\log p_{\hat{\boldsymbol{y}}}(\hat{\boldsymbol{y}})]$ is rate and $D = \mathbb{E}_{\boldsymbol{x}}d(\boldsymbol{x}, g_s(\hat{\boldsymbol{y}}))$ is distortion. $\lambda$ is a coefficient controlling the RD trade-off and $d$ is a distortion metric.

The vector quantization $Q$ is usually simplified to uniform quantization, *e.g.*,, rounding to the nearest integer, where $\hat{\boldsymbol{y}} = \lfloor \boldsymbol{y} \rceil = \boldsymbol{i}$. Most previous approximation methods are designed for uniform quantization. propose to add uniform noise on $\boldsymbol{y}$ during training. use straight-though estimator (STE) that copies gradient from $\hat{\boldsymbol{y}}$ to $\boldsymbol{y}$ to enable the training of encoder. Both and propose stochastic rounding that randomly quantizes $\boldsymbol{y}$ into two nearest integers, where anneals stochastic rounding to rounding during training. Agustsson & Theis (2020) propose a soft quantizer that smoothly interpolate between uniform noise and rounding. propose to optimize encoder with additive uniform noise and optimize decoder with rounding to reduce train-test mismatch. propose a two-stage strategy which first uses uniform noise for pre-training and then uses rounding for decoder finetuning. Among these methods, we can observe that additive uniform noise perform well for encoder

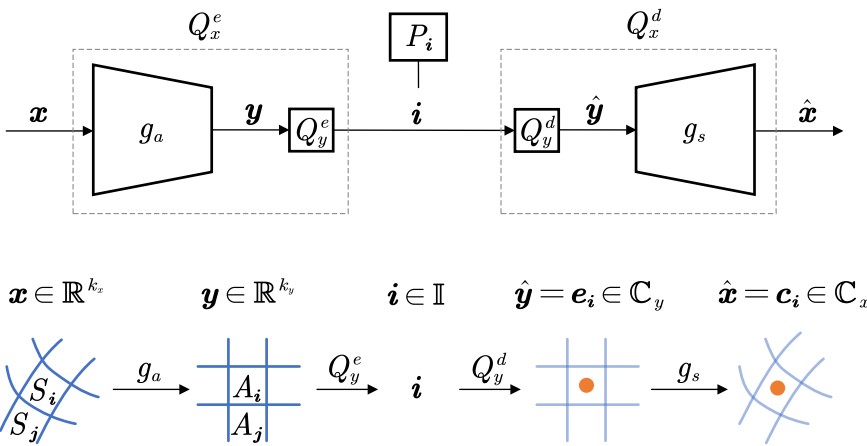

Figure 1: Interpreting neural compression as vector quantization. Blue lines are quantization boundaries and orange points are quantization centers.

optimization and rounding or annealing based rounding perform well for decoder optimization. In Section 3.2, we provide an explanation for this observation based gradient analysis.

For general vector quantization, the approximation design is more complicated. Agustsson et al. (2017a) propose a smooth approximation of vector quantization which is annealed to hard quantization during training. Zhu et al. (2022) replace vector quantization with a stochastic approximation that randomly quantize $\boldsymbol{y}$ to different codewords in the codebook. In VQVAE Van Den Oord et al. (2017), the authors use STE passing gradient from decoder to encoder, and introduce a VQ distance loss between $\boldsymbol{y}$ and codewords for the optimization of codebook and encoder. These method can only be optimized with distortion loss, where the rate is determined by codebook size. To achieve joint RD optimization, Feng et al. (2023) further improve the approximation in VQVAE with entropy-constrained vector quantization (ECVQ) Chou et al. (1989) in latent space. However, based on the analysis in Section 3.2, we argue that the encoder or decoder learned by previous methods are suboptimal in terms of RD performance.

## 3 GRADIENT ANALYSIS IN NEURAL COMPRESSION

In Section 2, we introduce the common architecture of neural compression and several VQ approximation methods. However, it is unclear about the impact of these approximation methods on the RD optimization. In the following, we first show that optimizing encoder and decoder is equivalent to optimizing the quantization boundaries and centers. Then we provide gradient analysis at boundaries and centers, showing the suboptimality of existing approximation methods of vector quantization.

### 3.1 INTERPRETING NEURAL COMPRESSION AS VECTOR QUANTIZATION

From VQ definition Gersho & Gray (1992), a vector quantizer $Q_x$ of size $N$ partitions the input vector space $\mathbb{R}^k$ into $N$ regions or cells. The region corresponding to the codeword $\boldsymbol{c}_i \in \mathbb{C}_x$ denote as $S_i$, where $i$ belongs to a index set $\mathbb{I}$ and $\mathbb{C}_x$ is the codebook. $S_i$ is defined as:

$$S_i = \{\boldsymbol{x} \in \mathbb{R}^k \mid Q_x(\boldsymbol{x}) = \boldsymbol{c}_i\} \tag{1}$$

Here, the regions satisfied $\bigcup_{i=1}^{N} S_i = \mathbb{R}^k$ and $S_i \cap S_j = \emptyset$ for all $i \neq j$. This implies that all the regions form a partition of the $k$-dimensional Euclidean space $\mathbb{R}^k$. The quantization boundaries are the partition boundaries, and the quantization centers are the codewords. The quantization boundaries and centers determine the RD performance.

For a neural compressor shown in Figure 1, we can regard the whole process $\hat{\boldsymbol{x}} = g_s \circ Q_y^d \circ Q_y^e \circ g_a(\boldsymbol{x})$ as a vector quantization process $\hat{\boldsymbol{x}} = Q_x(\boldsymbol{x})$ in source space. For the quantizer $Q_x$, the quantization

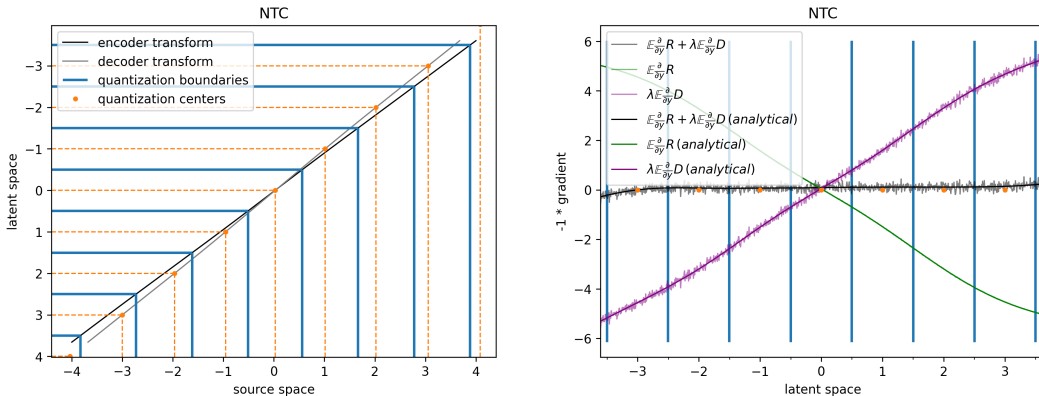

Figure 2: For UQ-AUN, the encoder-decoder mapping function (left) and the gradient with respect to $\boldsymbol{y}$ (right). Blue lines mark the quantization boundaries, and orange dots represent the quantization centers.

encoder is $Q_x^e = Q_y^e \circ g_a$ and the quantization decoder is $Q_x^d = g_s \circ Q_y^d$. It is important to know how a neural compressor determines the quantization boundaries and centers of $Q_x$ in source space.

In fact, the boundaries of $Q_x$ is determined by boundaries of $Q_y$ and encoder transform $g_a$; the centers of $Q_x$ is determined by centers of $Q_y$ and decoder transform $g_s$. Assuming input vector $\boldsymbol{x}$ is transformed into $\boldsymbol{y}$ and quantized to $\hat{\boldsymbol{y}} = \boldsymbol{e}_i \in \mathbb{C}_y$, where $\mathbb{C}_y$ is codebook of $Q_y$. We can define the latent space region $A_i$ partitioned by $Q_y$ as $A_i = \{\boldsymbol{y} \in \mathbb{R}^{k_y} \mid Q_y(\boldsymbol{y}) = \boldsymbol{e}_i\}$. As $\boldsymbol{y} = g_a(\boldsymbol{x})$ and $\hat{\boldsymbol{x}} = g_s(\hat{\boldsymbol{y}})$, we have:

$$S_i = \{\boldsymbol{x} \in \mathbb{R}^k \mid g_a(\boldsymbol{x}) \in A_i\} \tag{2}$$

$$\boldsymbol{c}_i = g_s(\boldsymbol{e}_i) \tag{3}$$

Since the quantization boundaries are uniquely determined by regions, the boundaries of $Q_x$ depend only on the encoder transform $g_a$ and the boundaries of $Q_y$. The quantization centers of $Q_x$ depend only on the decoder transform $g_s$ and the centers of $Q_y$. Moreover, if $\boldsymbol{y}$ lies on the boundary between two adjacent regions $A_i$ and $A_j$, then $\boldsymbol{x}$ will be on the boundary between $S_i$ and $S_j$. These finds show that optimizing the encoder and decoder is equivalent to optimizing the quantization boundaries and centers, providing insights on the design of approximation methods.

## 3.2 Gradient Analysis

During training, the quantized latent vector $\hat{\boldsymbol{y}}$ is replaced with a approximation $\tilde{\boldsymbol{y}}$, and $\hat{\boldsymbol{x}}$ is changed to $\tilde{\boldsymbol{x}} = g_s(\tilde{\boldsymbol{y}})$. With the per sample RD loss $l = -\log p_{\hat{\boldsymbol{y}}}(\hat{\boldsymbol{y}}) + \lambda d(\boldsymbol{x}, g_s(\hat{\boldsymbol{y}}))$, we care about the encoder gradient $\mathbb{E}\left[\partial l / \partial \boldsymbol{y}\right]$ and decoder gradient $\mathbb{E}\left[\partial l / \partial \tilde{\boldsymbol{x}}\right]$.

In fact, according to Section 3.1, if we fix the encoder (*i.e.*, fix the quantization boundaries) and optimize decoder with test-time quantization $\hat{\boldsymbol{y}}$, making the decoder gradient $\mathbb{E}\left[\partial l / \partial \hat{\boldsymbol{x}}\right]$ towards zero will lead to optimal optimization result of quantization centers. Therefore, when encoder is fixed, the best approximation $\tilde{\boldsymbol{y}}$ to optimize decoder is $\hat{\boldsymbol{y}}$ itself. In this section, we focus on analyzing the encoder gradient with different approximation methods for learning quantization boundaries.

**UQ-AUN** We start with uniform quantization for simplicity. Additive uniform noise (AUN) Ballé et al. (2017); Ballé et al. (2018a) is one of the most popular method to approximate uniform quantization during training. The rounding result $\hat{\boldsymbol{y}} = \lfloor \boldsymbol{y} \rceil$ is replaced with $\tilde{\boldsymbol{y}} = \boldsymbol{y} + \boldsymbol{u}$, where $\boldsymbol{u}$ is

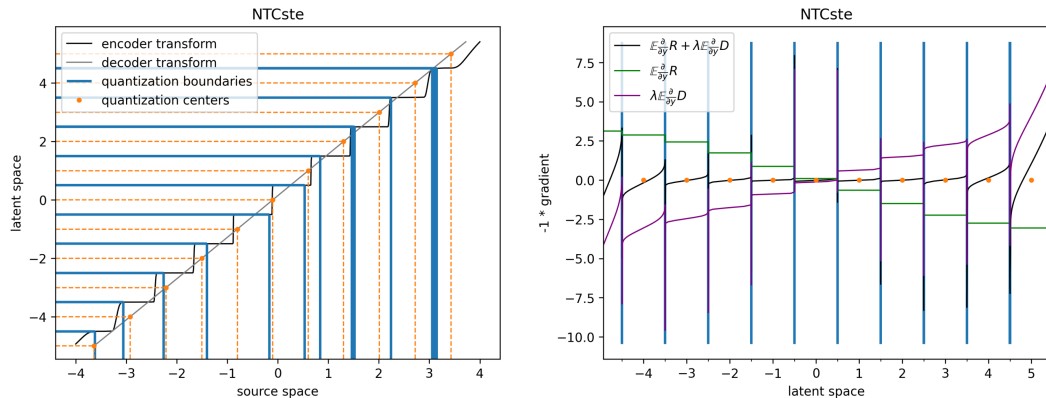

Figure 3: For UQ-STE, the encoder-decoder mapping function (left) and the gradient with respect to $\boldsymbol{y}$ (right). Blue lines mark the quantization boundaries, and orange dots represent the quantization centers.

sampled from uniform noise $U\left(\left[-\frac{1}{2}, \frac{1}{2}\right)^{k_y}\right)$. The encoder gradient of the scalar $y_1$ in $\boldsymbol{y}$ is:

$$
\begin{aligned}
&\mathbb{E}_{\boldsymbol{u}}\left[\frac{\partial l}{\partial \tilde{y}_1}\right] \\
&= \int_{y_1-0.5}^{y_1+0.5} \cdots \int_{y_k-0.5}^{y_k+0.5} \frac{\partial l(\tilde{\boldsymbol{y}})}{\partial \tilde{y}_1} \mathrm{d}\tilde{y}_1 \cdots \mathrm{d}\tilde{y}_k \\
&= \int_{y_2-0.5}^{y_2+0.5} \cdots \int_{y_k-0.5}^{y_k+0.5} l(y_1+0.5, \tilde{y}_2, \cdots, \tilde{y}_k) - l(y_1-0.5, \tilde{y}_2, \cdots, \tilde{y}_k)\mathrm{d}\tilde{y}_2 \cdots \mathrm{d}\tilde{y}_k
\end{aligned}
\tag{4}
$$

If $\tilde{y}_1$ is on the quantization boundaries, such as $y_1 = n + 0.5, n \in \mathbb{Z}$, the encoder gradient of $y_1$ is related to loss differences when $y_1$ is quantized to two nearby centers $n$ and $n+1$.

In the case of a one-dimensional source ($k = 1$), the encoder gradient is simplified to:

$$
l(y_1 + 0.5) - l(y_1 - 0.5)
\tag{5}
$$

In Section 3.1, we show that if $y$ lies at the boundary between two regions in the latent space, then $x$ is similarly positioned at the boundary of two corresponding regions in the source space. Consequently, when the gradient approaches zero at quantization boundaries, we have $l(n) = l(n + 1)$, which aligns perfectly with the boundary definition in optimal ECVQ Chou et al. (1989), given the quantization centers. In ECVQ, the loss of quantizing to two nearby centers is equal when $x$ is at the boundaries. This is why NTC Ballé et al. (2020) achieves near-optimal performance on 1-dimensional sources.

In Figure 2 (right), we illustrate the encoder gradients on a 1D Gaussian source. The gradients labeled "analytical" are calculated using Equation 4, while the unlabeled ones represent the averaged gradients over samples. The averaged gradients are smooth and closely match the theoretical results. Additionally, we show the encoder-decoder mapping function in Figure 2 (left). The encoder transform $g_a$ and decoder transform $g_s$ are not inverse functions of each other, leading to rate-constrained quantization results in source space (similar to ECVQ), where quantization boundaries are not at the center of two nearby quantization centers.

**UQ-STE** STE Bengio et al. (2013) is also an popular approximation. The value of $\tilde{\boldsymbol{y}}$ is the same as the value of $\hat{\boldsymbol{y}}$ but with modified gradient, where $d\tilde{\boldsymbol{y}}/d\boldsymbol{y} = 1$. We can represent it as $\tilde{\boldsymbol{y}} = \boldsymbol{y} + sg\left[\hat{\boldsymbol{y}} - \boldsymbol{y}\right]$, where $sg$ is the operation of stopping gradient. The encoder gradient for STE is equal to $\mathbb{E}\left[\partial l/\partial \hat{\boldsymbol{y}}\right]$. The gradient is discontinuous at boundaries because the $\hat{\boldsymbol{y}}$ suddenly changes from one quantization center to another one, as shown in Figure 3 (right). Moreover, the sum of the gradients on both sides of the boundary equals the difference in the derivatives of the RD loss, which can cause the RD optimization to get trapped in local optima. In Figure 3 (left), We show that the quantization boundaries optimized with UQ-STE are nonsmooth and suboptimal for RD performance.

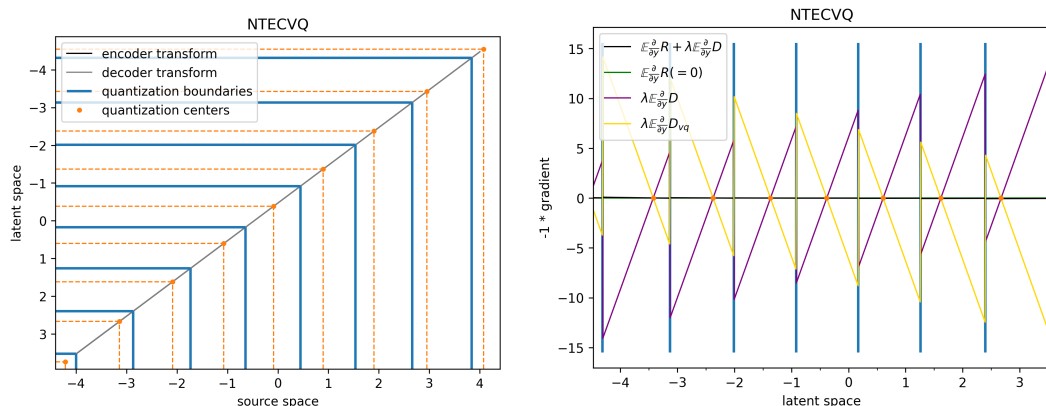

Figure 4: For VQ-STE, the encoder-decoder mapping function (left) and the gradient with respect to $\boldsymbol{y}$ (right). Blue lines mark the quantization boundaries, and orange dots represent the quantization centers.

**VQ-STE**   Since STE on uniform quantization does not define a way to optimize the codebook, it cannot be directly used in vector quantization with learnbale codebook. To simultaneously optimize the encoding transform and the codebook, VQVAE Van Den Oord et al. (2017); Razavi et al. (2019) introduce additional VQ distance loss $D_{vq}$ in latent space. The distance loss $D_{vq} = \mathbb{E}_{\boldsymbol{x}} d_1(\boldsymbol{y}, \boldsymbol{e}_i)$ is calculated between the latent vector $\boldsymbol{y}$ and the corresponding codeword $\boldsymbol{e}_i$, where $d_1$ is a VQ distance metric in latent space. To ensure end-to-end RD optimization, Feng et al. (2023) further introduce ECVQ and additional rate loss. The loss function is as:

$$L_1 = R + \lambda D + \beta D_{vq}, \tag{6}$$

where $\beta$ controls the trade-off between $d$ and $d_1$. Figure 4 illustrates that, unlike UQ-AUN, VQ-STE does not optimize the encoder based on the difference in RD loss. Instead, it optimizes the encoder by balancing the latent-space VQ distance loss $D_{vq}$ with the distortion loss $D$. This results in the encoder-decoder mapping becoming an identity mapping for 1D sources. The latent-space ECVQ in VQ-STE is equivalent to a source-space ECVQ. However, the issue of discontinuous gradients persists when $D_{vq}$ and $D$ cannot be properly balanced.

## 4   THE PROPOSED METHOD

### 4.1   ENCODER-DECODER ALTERNATING OPTIMIZATION

As shown in Figure 5, to address the train-test mismatch issues, this paper proposes an alternating optimization strategy for the encoder and decoder. When optimizing the quantization centers, the encoder is fixed, and the actual quantized values $\tilde{\boldsymbol{y}}$ are used to generate the reconstruction $\tilde{\boldsymbol{x}}$, after which the RD loss is computed to optimize the decoder and codebook. When optimizing the quantization boundaries, the decoder and codebook are fixed, and the quantization approximation $\tilde{\boldsymbol{y}}$ is used to generate the reconstruction $\tilde{\boldsymbol{x}}$, with the RD loss used to optimize the encoder. These two steps alternate during training. The entropy model is optimized during the first step.

### 4.2   STOCHASTIC VECTOR QUANTIZATION FOR ENCODER OPTIMIZATION

Consider a stochastic vector quantization, where the output $\tilde{\boldsymbol{y}}$ belongs to a conditional distribution $q(\tilde{\boldsymbol{y}} \mid \boldsymbol{y})$. We assume $d\tilde{\boldsymbol{y}}/d\boldsymbol{y} = 1$ and encoder gradient of the scalar $y_1$ in $\boldsymbol{y}$ is:

$$\mathbb{E}_{\tilde{\boldsymbol{y}}}\left[\frac{\partial l}{\partial y_1}\right] = \mathbb{E}_{\tilde{\boldsymbol{y}}}\left[\frac{\partial l}{\partial \tilde{y}_1}\right] = \int_{\omega} q(\tilde{\boldsymbol{y}} \mid \boldsymbol{y})\frac{\partial l}{\partial \tilde{y}_1}\,\mathrm{d}\tilde{y}_1 \mathrm{d}\tilde{y}_2 \cdots \mathrm{d}\tilde{y}_k \tag{7}$$

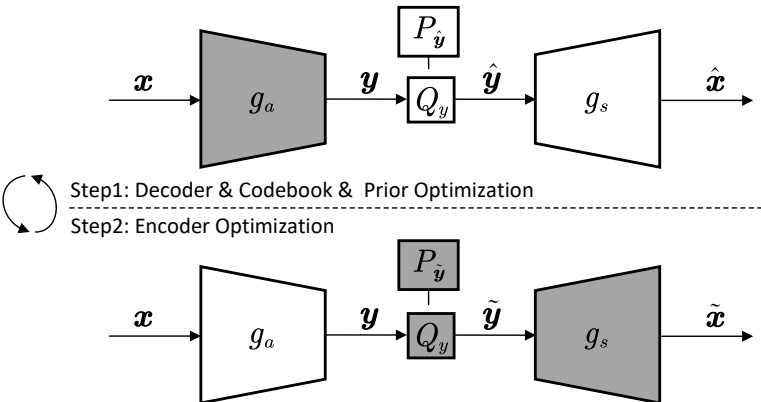

Figure 5: Alternating optimization of the encoder and decoder. Gray indicates freezed modules, while white indicates trainable modules.

$\omega$ is the integration area in $\mathbb{R}^k$. Let $q(\tilde{\boldsymbol{y}} \mid \boldsymbol{y})$ be a uniform sphere distribution centered at $\boldsymbol{y}$. The radius of the hypersphere is equal to $\|\boldsymbol{y} - \hat{\boldsymbol{y}}\|$. Therefore, the encoder gradient is as:

$$\int_{\omega} \frac{1}{V(\omega)} \frac{\partial l(\tilde{\boldsymbol{y}})}{\partial \tilde{y}_1} \, \mathrm{d}\tilde{y}_1 \mathrm{d}\tilde{y}_2 \cdots \mathrm{d}\tilde{y}_k, \tag{8}$$

where $V(\omega)$ is the volume of hypersphere, and $1/V(\omega)$ is the density because $\tilde{\boldsymbol{y}}$ is uniformly distributed. According to the generalized Stokes theorem, we have the gradient as:

$$\frac{1}{V(\omega)} \int_{\partial\omega} l(\tilde{\boldsymbol{y}}) \mathrm{d}\tilde{y}_2 \cdots \mathrm{d}\tilde{y}_k, \tag{9}$$

Therefore, the encoder gradient is the integration of loss function on the surface of the sphere. When $\boldsymbol{y})$ is at the boundaries, both the nearby two quantization centers $\boldsymbol{e}_i$ and $\boldsymbol{e}_j$ are on the surface, due to $\|\boldsymbol{y} - \hat{\boldsymbol{y}}\| = \|\boldsymbol{y} - \boldsymbol{e}_i\| = \|\boldsymbol{y} - \boldsymbol{e}_j\|$.

In fact, the proposed approximation is a generalization of additive uniform noise. If $q(\tilde{\boldsymbol{y}} \mid \boldsymbol{y})$ is uniform distributed within a unit hypercube centered at $\boldsymbol{y}$ with volume $V(\omega) = 1$, the gradient will be the same as that in Equation 4.

## 5 EXPERIMENTS

### 5.1 SETUP

**Source Data**  For vector sources, we conduct tests on 1-dimensional Gaussian sources, 2-dimensional Boomerang sources, and 8-dimensional Laplace sources. For natural image sources, we train on the train2017 dataset from COCO Lin et al. (2014), which contains 118,287 images. The training images are randomly cropped into 256×256 patches. The evaluation dataset is the Kodak dataset Kodak (1993), consisting of 24 images with a resolution of 768×512 pixels.

**Evaluation Metrics**  For vector sources, we use the following metric to measure distortion: $-10 \log(\mathrm{MSE}(\mathrm{x}, \hat{\mathrm{x}}))$, where MSE is the mean squared error. The bitrate is measured as bits per dimension (bpd). For natural images, the quality of the reconstructed images is evaluated using peak signal-to-noise ratio (PSNR) in the RGB color space, and the bitrate is assessed in bits per pixel (bpp). Both the distortion metrics $d$ and $d_1$ are mean squared error. Additionally, the BD-rate Bjontegaard (2001a) is employed to evaluate the average RD performance gain.

**Implementation Details**  For the model on low-dimensional vector sources, both the encoder and decoder transforms are constructed from Resblocks. The dimension of the latent-space vector is equal to the dimension of the source-space vector. For the model on image sources, the encoder and

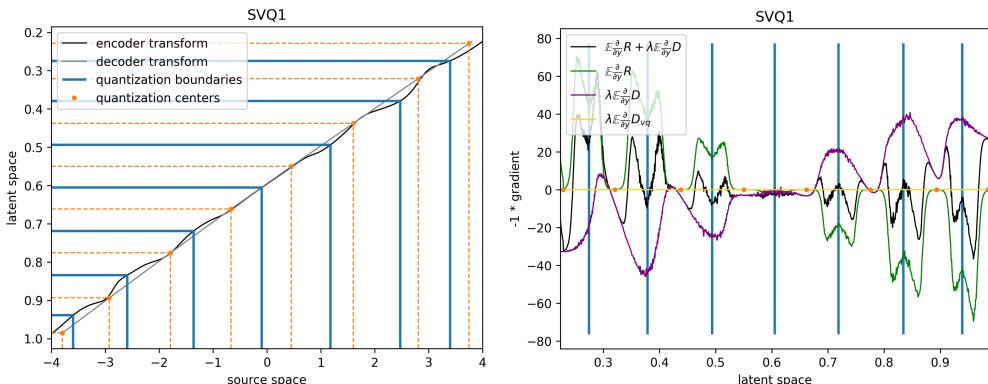

Figure 6: For the proposed method on 1D Gaussian source, the encoder-decoder mapping function (left) and the gradient with respect to $y$ (right). Blue lines mark the quantization boundaries, and orange dots represent the quantization centers.

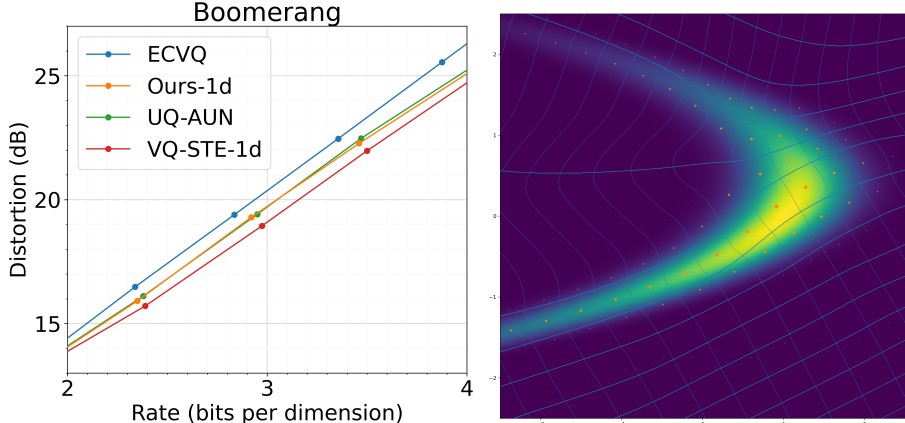

Figure 7: RD performance on the 2D Boomerang source (left) and the visualization of the quantization results of the proposed method (right).

decoder transforms follow the same structure as in the factorized model Ballé et al. (2018b), with the number of channels in the convolutional layers set to 192.

For the entropy model, we use the factorized entropy model Ballé et al. (2018b) when training with UQ-AUN and UQ-STE. When training with VQ-STE and the proposed method, we employ the discrete entropy model Van Den Oord et al. (2017); Feng et al. (2023), which consists of a softmax function and learnable logits.

For 1D, 2D, and 4D vector quantization, the codebook sizes are set to 256, 4096, and 32768, respectively. Since the codebook size required for vector quantization beyond 4 dimensions becomes excessively large without affecting performance, the experiments in this paper mainly focus on optimizing vector quantization for dimensions 4 and below.

We use the Adam optimizer Kingma & Ba (2014) for optimization, with a batch size of 1024 for low-dimensional vector sources and a batch size of 8 for image sources.

## 5.2 RESULTS AND ANALYSIS

In this section, we present the experimental results of the proposed method on different data sources, compare its performance with other methods, and conduct a series of ablation studies and analyses.

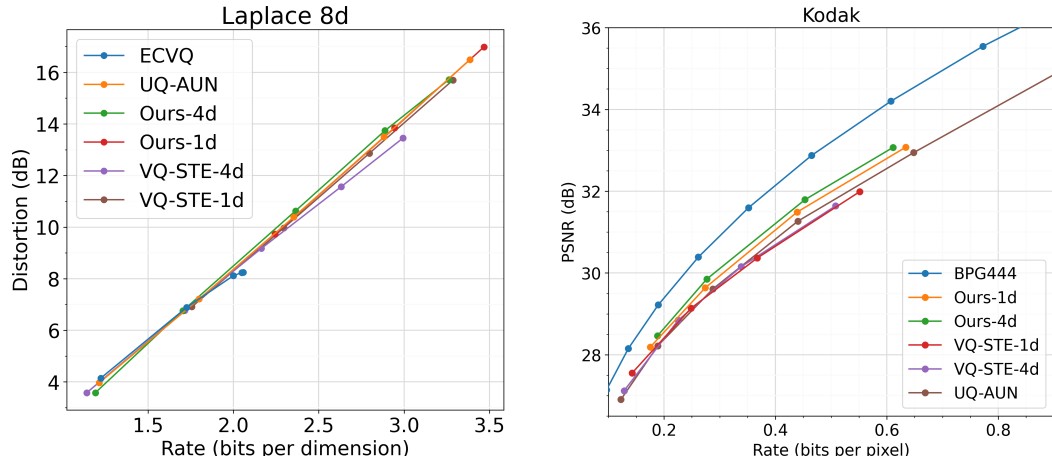

Figure 8: RD performance on the 8D Laplace source (left) and the Kodak image dataset (right).

Table 1: BD-rate comparison on Kodak dataset in terms of PSNR. The benchmark is UQ-AUN (Factorized model Ballé et al. (2018b)), with lower values indicating better performance.

| UQ-AUN | VQ-STE-1d | VQ-STE-2d | VQ-STE-4d | Ours-1d | Ours-2d | Ours-4d | BPG444 |
|--------|-----------|-----------|-----------|---------|---------|---------|--------|
| 0.0 | 1.15 | 2.42 | 0.16 | -5.58 | -7.20 | -9.39 | -26.16 |

### 5.2.1 LOW-DIMENSIONAL VECTOR SOURCES

**1D Gaussian Source**  For the 1D Gaussian source, the proposed method achieves performance very close to that of UQ-AUN. Here, we mainly showcase the encoder-decoder mapping function and the encoder gradient results for analysis. As shown in Figure 6, although the encoder gradients and encoder transform are not as smooth as those of UQ-AUN, the decoder transform remains sufficiently smooth and is able to learn nearly optimal mapping functions. Compared to UQ-AUN, the proposed method ensures train-test mismatch and is applicable to high-dimensional vector quantization. Additionally, compared to the UQ-STE method, the proposed method ensures optimal RD performance when determining quantization boundaries, avoiding the discontinuity in gradients.

**2D Boomerang Source**  For the 2D Boomerang source, we present the RD performance and the visualized quantization results of the proposed method. As shown in Figure 7, Ours-1d, UQ-AUN and VQ-STE-1d are there neural compressors, where the dimension of the latent-space vector quantizers is 1, *i.e.*,, scalar quantizers. The difference lies in that UQ-AUN uses uniform scalar quantization, while VQ-STE-1d and Ours-1d are scalar quantizers with learnable codebooks. It can be observed that VQ-STE-1d has a significant performance drop compared to UQ-AUN, the main reason for which is analyzed in Section 3.2. In contrast, the proposed method achieves results comparable to NTC.

**8D Laplace Source**  The experimental results on the 8D Laplace source are shown in Figure 8 (left). We performed both 1D and 4D vector quantization using the proposed optimization strategy. It can be observed that, even with scalar quantization, the performance of VQ-STE-1d, which uses the optimization strategy from previous work Feng et al. (2023), is slightly inferior to that of UQ-AUN. When the VQ dimension increases, the performance of VQ-STE-4d shows a significant drop. In contrast, the proposed method (Ours-1d) maintains performance on par with UQ-AUN in scalar quantization. As the quantization dimension increases to 4, Ours-4d shows improvements at higher bitrates, confirming its effectiveness. Notably, the performance of 8D ECVQ plateaus beyond 1.75 bpd due to its codebook size being insufficient to meet bitrate demands. At this rate point, the codebook size reaches 409,600. Due to the exponential growth in codebook size with increasing bitrates, further expansion becomes impractical.

### 5.2.2 NATURAL IMAGES

We also validate the effectiveness of the proposed method on the Kodak image dataset. Since the proposed alternating optimization strategy is only applicable to single-layer quantization and unconditional entropy models, we did not test on the state-of-the-art multi-layer quantization models for image compression. Instead, we tested the 1D, 2D, and 4D vector quantization results on the single-layer Factorized model Ballé et al. (2018b). The vector quantization is performed along the channel dimension. For example, in the case of 4D quantization, the $192 \times 1 \times 1$ channel vector is divided into 48 sub-vectors of size $4 \times 1 \times 1$, and vector quantization is performed on each sub-vector using 48 different codebooks, with the codebooks shared across the spatial domain. This quantization method only removes redundancy in the channel domain and does not address spatial redundancy.

The RD performance curve on the Kodak dataset is shown in the right column of Figure 8. Table 1 presents the BD-rate results Bjontegaard (2001b) with UQ-AUN as the baseline. It can be observed that the proposed method achieves steady performance improvements as the quantization dimension increases, while VQ-STE shows no significant improvement and even some performance degradation. Additionally, Ours-1d performs significantly better than UQ-AUN, primarily because alternating optimization resolves the train-test mismatch issue.

Table 2 presents a series of ablation experiments on the Kodak dataset. To verify the effectiveness of the alternating optimization (A1), we directly fed the quantization approximation results into the decoder and used an additional loss to constrain the learning of the codebook. However, the model without the alternating optimization strategy experienced training collapse, demonstrating the importance of alternating optimization for stable convergence.

Retaining the alternating optimization strategy, we replaced the proposed stochastic vector quantization method with two other approaches, including: soft-to-hard vector quantization Agustsson et al. (2017b) (A2), and probabilistic vector quantization Zhu et al. (2022) based on Gumbel Softmax Maddison et al. (2017) (A3). The rate of these methods is controlled by adjusting the codebook size. It can be observed that, with the same transform structures and optimization strategy, the proposed sphere-noise based stochastic approximation achieves better RD performance compared to other VQ approximation.

Table 2: Abaltion studies on Kodak dataset in terms of PSNR. The benchmark is UQ-AUN (Factorized model Ballé et al. (2018b)), with lower values indicating better performance.

|  | BD-rate |
| --- | --- |
| UQ-AUN | 0.0 |
| Ours-4d | -9.39 |
| A1: Ours-4d w/o alternating optimization | NaN |
| A2: Ours-4d + Agustsson et al. (2017b) | 25.31 |
| A3: Ours-4d + Zhu et al. (2022) | 16.25 |

## 6 CONCLUSION

In this paper, we propose a method named Alternating Optimized Stochastic Vector Quantization to address the RD optimization issue in vector quantization based neural compression. We propose an encode-decoder alternating optimization strategy. The encoder is optimized with differentiable approximation, and the decoder is optimized with actual quantization to avoid the train-test mismatch of quantization centers. For better encoder optimization, we propose a sphere-noise based stochastic approximation method. During encoder optimization, VQ is replaced with a uniform sphere noise centered at the input vector. When the input vector is located at the quantization boundary, the encoder gradient is closer to the difference in RD loss between adjacent quantization centers, facilitating better encoder optimization. We provide a thorough analysis using toy vector sources and demonstrate through extensive experiments on neural image compression that our proposed method achieves a significant performance gain.

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

## A  APPENDIX

You may include other additional sections here.

