# OpenReview forum: "Alternating Optimized Stochastic Vector Quantization in Neural Compression"
_ICLR.cc/2025/Conference — ICLR 2025 Conference Withdrawn Submission_

### Official Review · Reviewer_X4Nj · 2024-10-29

**Soundness:** 3
**Presentation:** 3
**Contribution:** 2
**Rating:** 5
**Confidence:** 4

**Summary:**

This paper proposes an alternating optimization method that incorporates stochastic quantization to improve the quantization process of nonlinear transform coding (NTC). The paper clearly formulates the optimization problem of NTC from the perspective of vector quantization, *i.e.*, the optimization of boundaries and codewords. Experiments on low-dimensional sources and natural images show that the proposed method outperforms the classical NTC method equipped with additive uniform noise and straight-through estimator on image compression.

**Strengths:**

1. The paper is overall well written and easy to follow.
2. The authors provide a clear framework for analyzing the gradient approximation problem of NTC and propose a method for solving it based on the characteristics of vector quantization.

**Weaknesses:**

1. The motivations and advantages of employing a uniform sphere distribution are hard to understand. The uniform quantizer with additive uniform noise also approximates the encoder gradient with the difference in RD loss between adjacent quantization centers (which is the main advantage of the uniform sphere distribution), as shown in Eq. (4).

   By the way, I noticed that the proposed method uses a learnable multidimensional codebook instead of a fixed codebook of uniform quantizers. However, such a gap can be reduced by the nonlinear transforms (for flexible boundaries and codebook in the source space) and conditional coding (for redundant multidimensional signals).

2. The importance of the proposed method seems to be limited. Vector quantization and conditional coding (*e.g.*, spatial auto-regression [R1] and channel-wise auto-regression [R2]) are two kinds of methods that solve the high-dimensional coding problem of latent representations, and the latter one is more prevalent in existing methods. Theoretically, the proposed alternating method can be used in both vector quantization and conditional coding. However, the authors only offer the results for vector quantization. It is better to evaluate the contribution of the proposed method by integrating it with state-of-the-art conditional coding methods, such as ELIC [R3] and TCM [R4].

   [R1] D. Minnen, J. Ballé, and G. D. Toderici. Joint autoregressive and hierarchical priors for learned image compression, In *Advances in Neural Information Processing Systems (NeurIPS) 31*, 2018, pp. 10771-10780.

   [R2] D. Minnen and S. Singh. Channel-wise autoregressive entropy models for learned image compression. In *2020 IEEE International Conference on Image Processing (ICIP)*, 2020, pp. 3339-3343.

   [R3] D. He, *et al.* ELIC: Efficient learned image compression with unevenly grouped space-channel contextual adaptive coding. *Proceedings of the IEEE/CVF Conference on Computer Vision and Pattern Recognition (CVPR)*, 2022, pp. 5718-5727.

   [R4] J. Liu, H. Sun, and J. Katto. Learned image compression with mixed transformer-cnn architectures. *Proceedings of the IEEE/CVF conference on computer vision and pattern recognition (CVPR)*, 2023, pp. 14388-14397.

3. Contributions on interpreting neural compression as vector quantization should be clarified. There has been work (Ballé *et al.*, 2020) that reveals the relationship between the source domain and the latent representation. Although this paper is cited by the authors in their related work, the relationship and contributions of the two papers are not clarified.

4. Several details should be clarified in the manuscript to ensure that the paper is self-contained.

   - The implementation of vector quantization in the latent space, which is crucial to better understand the contribution of the proposed method.

   - The definition on the uniform sphere distribution.

     I note that there are two different definitions of hypersphere, with a difference in whether the points with a distance less than the radius are considered part of the hypersphere. It is suggested that the authors provide a clear definition.

     (Additional) 2 definitions, with the latter one be the case of this paper:

     a) The $(k-1)$-sphere with a radius $R$ is the set of points $[x_1, x_2, \cdots, x_k]$ with $\sum_{i=1}^kx_i^2 = R^2$.

     b) The $k$-dimensional hypersphere with a radius $R$ is the set of points $[x_1, x_2, \cdots, x_k]$ with $\sum_{i=1}^kx_i^2\leqslant R^2$.

5. Typos:

   - There are several omitted citations in the second paragraph of Section 2.

   - There is a redundant comma after “e.g.,” in Line 99.

   - The references are not cited with proper commands. Some of the citations need to be replaced by `\citep` instead of `\citet`.

   - There is an unnecessary bracket after $\mathbf{\mathit{y}}$ in Line 353.

**Questions:**

1. What is the main advantage of using noise that follows a uniform spherical distribution over conventional additive uniform noise?
2. In the figures, the encoder transform of UQ-STE (Figure 3) includes the quantizer while that of UQ-AUN (Figure 2) does not. Why?
3. What's the definition of hypersphere in this paper?
4. Why the prior is optimized together with the decoder instead of the encoder in the alternate optimization? The distribution of the codeword is determined only by the boundaries, which are determined by the encoder.
5. How to guarantee $\Vert\mathbf{\mathit{y}}-\hat{\mathbf{\mathit{y}}}\Vert= \Vert\mathbf{\mathit{y}}-\mathbf{\mathit{e}}_i\Vert=\Vert\mathbf{\mathit{y}}-\mathbf{\mathit{e}}_j\Vert$ for the vector quantizer?
6. Is the proposed alternating optimization method applicable to other NTC models, including those with uniform quantizers?

---

### Official Review · Reviewer_udHP · 2024-11-04

**Soundness:** 2
**Presentation:** 3
**Contribution:** 2
**Rating:** 3
**Confidence:** 5

**Summary:**

The paper addresses two main issues of vector quantization (VQ) approximation methods in neural compression. The paper proposes encoder-decoder alternating optimization strategy to address the train-test mismatch and stochastic sphere-noise based approximation technique for suboptimal encoder gradients for rate-distortion (R-D) optimization. Experimental results on synthetic sources and natural images demonstrate the effectiveness of the proposed method over previous VQ approximation methods in terms of R-D performance.

**Strengths:**

1. The paper is well written and easy to follow.

2. The proposed stochastic vector quantization for encoder optimization approach is superior to the previous VQ+STE approximation method as well as the UQ+AUN method, as demonstrated in experiments.

**Weaknesses:**

1. The proposed encoder-decoder alternating optimization strategy is of less importance. Recent neural compression methods address the train-test mismatch issue in end-to-end training by adopting mixed quantization. That is using additive uniform noise for learning the entropy model but employing quantized latent when it is passed to the decoder. There is no evidence that the encoder-decoder alternating optimization strategy is better than the mixed quantization method. Moreover, as the authors illustrated, the proposed alternating optimization strategy is only applicable to single-layer quantization and unconditional entropy models, which leads to obviously degraded R-D performance.

2. In the proposed stochastic vector quantization approach, the authors assume $q(\tilde{y}|y)$ is a uniform sphere distribution centered at $y$. However, there is no theoretical evidence to support that this assumption is reasonable.

3. In experiments:

(1) For low-dimensional vector sources, it is not reasonable for the dimension of the latent-space vector to be the same as that of the source-space vector, as the primary task of the encoder is dimensionality reduction for feature extraction .

(2) The specific structure of the entropy model of VQ-STE and the proposed method is not given. Due to the different entropy models, it is also unfair to compare the proposed method with UQ-AUN and UQ-STE.

(3) The R-D performance of the proposed method is evidently worse than current state-of-the-art methods. It is even worse than BPG444.

**Questions:**

1. What are the advantages of the proposed encoder-decoder alternating optimization strategy over mixed quantization method?

2. Could the authors theoretically prove that the assumption of $q(\tilde{y}|y)$ being a uniform sphere distribution centered at $y$ is valid?

3. Could the performance of the proposed model achieve state-of-the-art results？

---

### Official Review · Reviewer_GzUa · 2024-11-08

**Soundness:** 2
**Presentation:** 1
**Contribution:** 2
**Rating:** 3
**Confidence:** 4

**Summary:**

In this paper, the authors propose an optimization strategy for vector quantization in neural compression. Since quantization is non-differentiable, they approximate the vector quantization error using noise sampled from a uniform spherical noise distribution. Additionally, they introduce an optimization strategy to effectively minimize the rate-distortion loss function in neural compression. The authors tested their method on simulated data sources and several real-world images, demonstrating that their approach provides better compression efficiency compared to existing vector quantization methods.

**Strengths:**

1. An alternative optimization procedure to optimize the encoder network, the codebook of vector quantization, and the decoder network. This procedure could result in better convergence of the RD loss function.
2. An approximation of vector quantization using uniform spherical noise centered on the latent vector.
3. A gradient analysis of the encoder latent with respect to the loss function.
4. Deriving the correspondence between vector quantization in the latent space and the corresponding quantization in the image space.

**Weaknesses:**

1.The paper is not well-written and is incomplete in several sections. In the related work section, citations are missing, and sentences are incomplete, making it difficult to relate the written content to the prior art. Few of the papers in the reference are repeated.

2. The evaluation of the proposed vector quantization is limited. The authors have only experimented with a low-complexity autoencoder using a single layer. Consequently, the impact of the proposed method on neural compression is limited. The authors should utilize recent state-of-the-art variational autoencoder-based neural image compression methods, such as [1] and [2], and apply the proposed vector quantization to the latent space of these advanced methods. When the encoder and decoder are more powerful, the impact of vector quantization on reducing the bitrate might be lower than what is shown in the paper.
 [1] Cheng et. al, Learned Image Compression with Discretized Gaussian Mixture Likelihoods and Attention Modules, CVPR 2020
 [2] He et.al, ELIC: Efficient Learned Image Compression with Unevenly Grouped Space-Channel Contextual Adaptive Coding, CVPR 2022.

3. The details of the network architecture are missing from the paper.

4. The alternative optimization strategy is well-established in the vector quantization literature, where the codebook is fixed while optimizing the encoder and decoder. Additionally, in neural compression, some prior works [3] perform fine-tuning of the decoder using the quantized latent \hat{y}​, showing that optimizing the decoder with the quantized latent improves compression efficiency and reduces the train-test set mismatch. The citations are missing.
   [3] Zongyu Guo et.al, Soft then Hard: Rethinking the Quantization in Neural Image Compression, ICML 2021

5. The citations to the related work (baseline) are incorrect (e.g., in Table 1), making it difficult to review the paper.

**Questions:**

1. What is the single-layer factorized model? Is it the encoder with a single layer, or is it the factorized entropy model with a single layer? The description of the network architecture is not clear in the paper.

2. Please provide more details on the optimization of quantization boundaries. When the codebook is fixed, the decoder network and the entropy model are fixed, and the quantization boundaries depend on the codebook centers. How are the boundaries defined? Is it with respect to nearest-neighbor-based partitioning? When the encoder is optimized, the encoder might move the latent into a different partition. Is this what is meant by the optimization of quantization boundaries?

3. The rate of the baseline methods is controlled by adjusting the codebook sizes. Why is the entropy model not used for the baseline methods in the comparison? Even though the baseline methods do not consist of the entropy model, it is better to include the entropy model. The BD-rate gain for the proposed method could also come from the use of the entropy model, in addition to the proposed vector quantization method. The baseline method with the entropy model might also have similar results to the proposed method. If the baseline method also includes the entropy model, it will be easier to quantify the improvement of the proposed vector quantization.

4. In Table 1, for the baseline UQ-AUN (Factorized model Balle et al. (2018b)), is the hyper-prior entropy model used, or is the citation incorrect? In the text, it is written as the factorized entropy model, but it is cited with the hyper-prior entropy model: Johannes Balle, David Minnen, Saurabh Singh, Sung Jin Hwang, and Nick Johnston. Variational image compression with a scale hyperprior. arXiv preprint arXiv:1802.01436, 2018b.

---

### Official Review · Reviewer_am93 · 2024-11-08

**Soundness:** 3
**Presentation:** 2
**Contribution:** 2
**Rating:** 3
**Confidence:** 4

**Summary:**

This paper investigates an improvement to the STE method used to train VQ-based neural compressors. For scalar quantization methods, the uniform additive noise method during training is shown to yield smooth gradients. This is not applicable to VQ-based methods, which so far mostly use STE. This is shown to yield highly non-smooth gradients. The proposed method, for VQ-based models, uses an alternating optimization scheme, combined with stochastic VQ. This is shown to yield smoother gradients than STE. Experimental results demonstrate superiority over STE-based VQ neural compressors.

**Strengths:**

- The problem of train-test mismatch and other issues of STE in VQ-based models is relevant and timely
- The proposed method appears principled, and solves some of the challenges that are presented
- The work is overall well-motivated, and easy to follow

**Weaknesses:**

- In sections 1-2, the problem is presented well, i.e., the need to solve some issues brought forth by STE in VQ-based compressors. However, section 3 dedicates a lot of explanation to how it is solved in scalar quantized neural compressors, which, to me, appears less important. In 3.2, I think it would be helpful to directly mention the VQ-STE section, as that is the setting which this paper's proposed method attempts to improve on. The UQ-AUN and UQ-STE can be mentioned briefly and details put in the appendix, as the scalar quantization setting is not the focus of the paper. This would provide more space to explain some of the details of the proposed method in section 4, which I found to be lacking. In addition, Figure 6 could be placed in section 4, and the reader can directly contrast that with Figure 4, and see how the non-smoothness issue is fixed via the proposed method.
- The experimental results section covers a broad range of sources, both synthetic and real-world, which is helpful. It is shown that the proposed method outperforms VQ-STE in all settings, and the UQ-AUN method provides a frame of reference. However, some baselines are missing. For example, the two methods soft-toward vector quantization (A2) and probabilistic vector quantization (A3) used in the ablation study (lines 509-511) should also be its own baselines with the Balle et al 2018 transforms. This is useful for understanding how the proposed method compares with other methods that don't use STE. Moreover, these baselines are mentioned in the related work but not compared to.
- In the related work, lines 138-140, it is said that section 3.2 addresses how prior works in VQ-based neural compression yield sub optimality. However, in the VQ setting, only the STE method from VQVAE is addressed. The method from Agustsson et al, 2017, and Zhu et al 2022 are not addressed in section 3.2. It would be helpful to understand how these two methods' gradients look like in the 1-d Gaussian setting. This, combined with a BD-rate comparison in the results section, would help the reader understand how all the methods compare (conceptually and performance-wise), and strengthen the work overall.
- Furthermore, the experimental results of the proposed method on natural images use a fairly old architecture (which, to my understanding, uses transforms from Balle et al 2018, single-layer vector quantizer, and a discrete entropy model from VQVAE). There are more recent transforms that are higher-performing, such as those from [1], as well as vector quantizer layers, such as those from [2] and [3]. Experiments using these models would be more convincing. The authors say the proposed method cannot be used on more state-of-the-art models such as these. If true, I think that limits the applicability of the proposed method.
- There are some issues with the references in the related work, in the second paragraph.

References:

[1] Cheng, Zhengxue, et al. "Learned image compression with discretized gaussian mixture likelihoods and attention modules." Proceedings of the IEEE/CVF conference on computer vision and pattern recognition. 2020.

[2] El-Nouby, Alaaeldin, et al. "Image compression with product quantized masked image modeling." arXiv preprint arXiv:2212.07372 (2022).

[3] Feng, R., Guo, Z., Li, W., & Chen, Z. (2023). NVTC: Nonlinear vector transform coding. In Proceedings of the IEEE/CVF Conference on Computer Vision and Pattern Recognition (pp. 6101-6110).

**Questions:**

- For the natural image setting with the proposed method, are the transforms from Balle et al 2018, and entropy model the discrete entropy model from VQVAE?
- Why can the proposed method not be applied to architectures like NVTC [1] or PQ-VAE [2]? This is not explained, and it seems like the proposed method could be used on these architectures.

References:

[1] El-Nouby, Alaaeldin, et al. "Image compression with product quantized masked image modeling." arXiv preprint arXiv:2212.07372 (2022).

[2] Feng, R., Guo, Z., Li, W., & Chen, Z. (2023). NVTC: Nonlinear vector transform coding. In Proceedings of the IEEE/CVF Conference on Computer Vision and Pattern Recognition (pp. 6101-6110).

---

### Note · Authors · 2024-11-15

**Comment:**

We gratefully thanks for the reviewer's comments and suggestions.

Although the current version of the paper has many weaknesses, we would like to emphasize that :
(1) the alternative optimization strategy is important to reduce train-test mismatch in decoder and codebook optimization.
(2) the sphere-noise based approximation is important to provide a smoother and more optimal gradient for encoder optimization.

In the future, we plan to revise the paper as follows:
(1) We will extend our method to multi-layer quantization architectures (e.g., hyperprior model, ELIC, NVTC, etc.). The quantization boundaries and centers in these models are more complex, which will require additional effort in method design.
(2) We will reorganize the structure of the paper and provide more detailed gradient analysis for other VQ-based approximation methods.
(3) We will conduct an ablation study on applying the alternative optimization strategy to existing NTC methods with scalar quantization and compare it with mixed quantization strategies and decoder fine-tuning strategies.
(4) We will include more implementation details, such as network structures and descriptions of the entropy model.
(5) We will revise any typographical errors.

Thank you once again for taking the time to review

**Withdrawal Confirmation:**

I have read and agree with the venue's withdrawal policy on behalf of myself and my co-authors.